# Quality Improvement of the Forging Process Using Pareto Analysis and 8D Methodology in Automotive Manufacturing: A Case Study

**Katarína Lestyánszka Škůrková [1],\*, Helena Fidlerová [1] , Marta Niciejewska [2] and Adam Idzikowski [2]**

[1] Faculty of Materials Science and Technology in Trnava, Institute of Industrial Engineering and Management, Slovak University of Technology in Bratislava, 91724 Trnava, Slovakia; helena.fidlerova@stuba.sk

[2] The Faculty of Management, Department of Production Engineering and Safety, Czestochowa University of Technology, 42-201 Czestochowa, Poland; marta.niciejewska@wz.pcz.pl (M.N.); adam.idzikowski@wz.pcz.pl (A.I.)

\* Correspondence: katarina.skurkova@stuba.sk

**Abstract:** A systematic approach to nonconformity management and continuous improvements are the key elements of the quality management system. The objective of this paper is to present quality improvement for a manufacturing company producing forgings through the combination of several statistical methods and quality control techniques. First, Pareto analysis was applied, followed by the 8D (eight disciplines) methodology using a structured eight-step approach to problem solving following the PDCA (plan-do-check-act) method. The aim was to identify the problem and discover the critical causes of failures in the management system that allowed the problem to occur, by modifying and combining appropriate quality management methods and tools. The paper introduces a case study considering the forging process in the production of gearboxes, where customer complaints were identified in the last year, demanding the need to eliminate failures. Using the mentioned methods, the root cause of the problem was identified and permanent corrective action was planned and implemented according to the recommendations of the 8D report, which made it possible to reduce the likelihood of a recurrence of this problem and increase customer satisfaction.

**Keywords:** quality control; complaints; 8D report; forging process; gearbox; problem solving; Pareto analysis; case study





## 1. Introduction

Quality requirements have expanded to the point where quality management has become the decisive factor in company management. Implementing quality standards allows for the maintenance of one's position in the market, the gain of a competitive advantage, and the ability to compete in the sector [1].

Global 8D comprises a problem-solving methodology that seeks to identify and eliminate root causes of failure after the fact. The Global 8D problem-solving methodology was originated by the US government following World War II as MIL-STD 1520 and was later adopted and improved upon by the Ford Motor Company. Failure mode and effects analysis (FMEA) attempts to anticipate and correct problems before they occur [2,3]. Global 8D deals with the modes and causes of failure after the fact. Once used, Global 8D results can help improve the quality of future FMEA of a product, process, or service [4].

Global 8D is a methodology to identify and eliminate the root causes of failure and then implement permanent corrective actions to prevent a recurrence. The method is also designed to evaluate the effectiveness of failure mechanism controls that allowed the problem to escape undetected in the first place [4,5]. The method is typically used within an organization, and its results can be presented to the customer as quality control [6,7].

The output of the 8D method is an 8D report based on the multi-stage work of the entire team using effective methods and tools for quality management or improvement [8,9].

The primary objective of the 8D methodology is to implement and consolidate corrective actions in relation to the quality management system. It comprises eight stages that set out a procedure to follow an established pattern [3,7,9]. Each of them must be recorded in a document called the 8D Report [10].

The new benefit identified in this case study for quality control, not mentioned in previous studies [10,11], is the combination of many tools of quality management, especially Pareto analysis, RE analysis, and the five whys technique. As a result, a synergistic process is developed. The quality of the process has been improved, because the root causes of problems have been identified and removed.

## 2. Materials and Methods

*Research Aim and Methodology*

The aim of the study is to consider quality improvement in industry to meet customer expectations. We assume that this study can be an example of the specifics of applying quality management tools to meet the stated goal. In the introductory part of the research, we focused on choosing the right approach and method by studying the literature. An important task was not only to identify errors but also to focus on the causes of errors. Another reason for choosing 8D was that it ensures, as mentioned, the effectiveness of failure mechanism controls, while consolidating their use in the future. The findings and experience can be used as a basis for future FMEA analysis of an automotive product.

For analysis, a monitoring and improvement project was chosen for the most demanded product of the forging process. As the most appropriate method, the 8D methodology has been applied. The study was conducted in an organization in western Slovakia dealing with forging and steel processing. Customers' requirements for quality in automotive production are very strict. The project's issue has been a desire to improve the forging process by using the 8D methodology. The Global 8D problem-solving methodology was originated by the U.S. government following World War II as MIL-STD 1520, and was later adopted and improved upon by the Ford Motor Company [4]. The 8D methodology was used to analyze the problem, which allowed for the identification of a critical cause of the defect and helped eliminate it. The 8D methodology includes an 8D report consisting of eight steps, as shown in Figure 1:

- 1D: Team Formation.
- 2D: Problem Definition and Description.
- 3D: Interim Containment Actions.
- 4D: Root Cause and Effect Analysis.
- 5D: Corrective Actions.
- 6D: Verification of Corrective Actions.
- 7D: Preventive Actions.
- 8D: Team and Individual Recognition.

The primary objective of the 8D methodology is to implement and consolidate corrective actions in the quality management system. It comprises eight stages that set out a procedure to follow an established pattern.

Global 8D employs a gateway step (D0) followed by eight discipline steps (D1–D8) [3,9].

Gateway step discipline D0—Prepare for Global 8D. This step confirms that the Global 8D methodology is needed. Symptoms are documented, showing that a problem has occurred, and a formal assessment is performed to justify the deployment of the resources required to perform a team-based cross-disciplinary problem-solving effort. Immediate damage control is conducted in the form of emergency response actions to prevent further undesirable consequences.

Discipline D1—Create an interdisciplinary problem-solving team. The problem-solving team should consist of individuals who have the knowledge, time, authority, and skills to solve the problem and follow through by implementing corrective actions.

The team composition should be cross-functional and include representation from all areas that need to be involved in containing, analyzing, correcting, and preventing the problem at hand.

Discipline D2—Define the problem. The team might describe the problem as specific as possible, including such details as: who suffers if the problem goes unsolved; what is the cost of not solving the problem; when was the problem discovered, how was it found, where was it found, and by whom; a description of the failure mode and rate; and any metrics and measures relevant to the problem situation.

Discipline D3—Contain the problem. In this discipline, the team focuses on containing the issue. Intermediate containment actions are developed and implemented. Measures are taken to protect customers from the problem until permanent corrective actions can be taken. For example, all production that has been potentially affected by the problem should be isolated for detailed inspection. Those defective products that have been shipped may need to be recalled, and the production of those still in the manufacturing facility should be placed on hold until the problem has been fixed. The team needs to verify the effectiveness of these actions.

Discipline D4—Identify root causes and escape points. During the fourth discipline, the problem-solving team identifies all potential causes and gathers as much evidence as possible to reliably test each potential cause against the problem data. When a cause-and-effect relationship is established, a detailed description of how the cause led to the failure is formulated. At this point, an escape point is identified where the control mechanism broke down and allowed the problem to go undetected. The escape point is identified as the first control point following the root cause. This constitutes the earliest opportunity to expose the problem if the control point had been effective. Once all root causes and escape points have been identified, the team can generate possible alternatives for corrective action.

Discipline D5—Develop permanent corrective actions (PCAs). Discipline D5 is concerned with choosing permanent corrective actions and documenting the rationale for each. The team confirms that the recommended PCAs will solve the problem and will not produce any negative consequences. The implementation of PCAs can require a preliminary evaluation and, in some cases, a small pilot study. During this step, PCAs must also address control issues posed by the escape point.

Discipline D6—Implement permanent corrective actions. In discipline six, the team implements the PCAs identified in discipline five. Data are collected showing that corrective actions effectively prevent a recurrence of the root cause. This includes a demonstration of how the escape point control mechanism has been improved in its capability for early detection.

Discipline D7—Prevent recurrence. Preventing recurrence of the problem requires expanding the scope of PCAs and controls to apply to other similar products, processes, or services. The standardization and deployment of corrective actions across all products or services that might be subject to the same or similar problem leverage the problem-solving effort, becoming a preventative and proactive measure across the production facility.

Discipline D8—Give the team credit. The last step of the Global 8D process is formally recognizing the collective efforts of the problem-solving team and formally approving its report. Achievements should be widely promoted, and the acquired knowledge and learning should be freely shared.

The most important aspect of quality control in this case study is the combination of many tools of quality management, especially Pareto analysis, RE analysis, and the five whys technique. We consider the implementation of the 8D method to be extremely important for improving quality and meeting the needs of the customer in the automotive sector. By selecting and modifying this method, the needs of the specific company for the selected product are addressed, and it is possible to make the production process more efficient.

Finally, we also attached recommendations for modification and further use of 8D in industry.

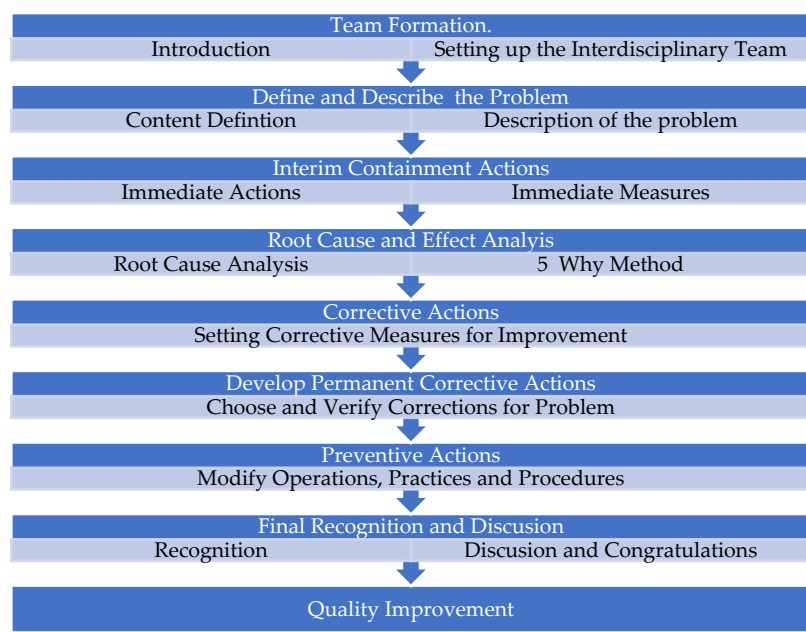

**Figure 1.** The 8 D methodology workflow.

### 3. Results

The study was conducted in a company producing forgings and steel processing. As mentioned above, the quality requirements of automotive production customers are very strict. As stated by the authors [12], large steel products produced by forging processes are widely used in industries with high-quality demands on structural and mechanical properties [12]. The monitored product is the forging for the gearbox. First, statistical methods of quality control with descriptive characteristics as relative and absolute frequencies are used for implementation of the Pareto principle to identify critical failures. Using statistical process control, organizations can optimize manufacturing processes, ensuring better product quality, better understanding of the relationships between process inputs and outputs, lower costs due to the elimination of nonconforming products, and increased profits [13]. Then the 8D report is implemented by an expert cross-functional team.

#### 3.1. Pareto Analysis

First, Pareto analysis was used to identify and classify the most common failures in forging production (Figure 2). Pareto analysis does not belong to a particular single step of 8D methodology, but this tool helps us to find out which problem is the most important of all. This problem will then be solved by 8D methodology. The Pareto principle says that 80% of consequences come from 20% of causes called the vital few. It helps us identify critical failures in production [13].

Possible problems were identified by analyzing customer complaint records, and their classification with decreasing frequency of occurrence are as follows: wrong size, bad forging, technological waste, machine failure, material fallen out, badly cut forging, burned out, and needle pushed. The frequencies, absolute frequencies, and cumulative absolute frequencies are calculated using statistical methods for further analysis. The frequency of the defects used then for the Pareto analysis chart was compiled based on the calculated data above.

As shown in Figure 2, the most frequent mistakes are identified as follows: wrong size, bad forging, and technological waste with nearly 80% cumulative frequency. Based on these findings, we focus on the steps of analysis using the 8D method and report on the most frequent mistakes that occur in the production of forgings identified as "wrong size".

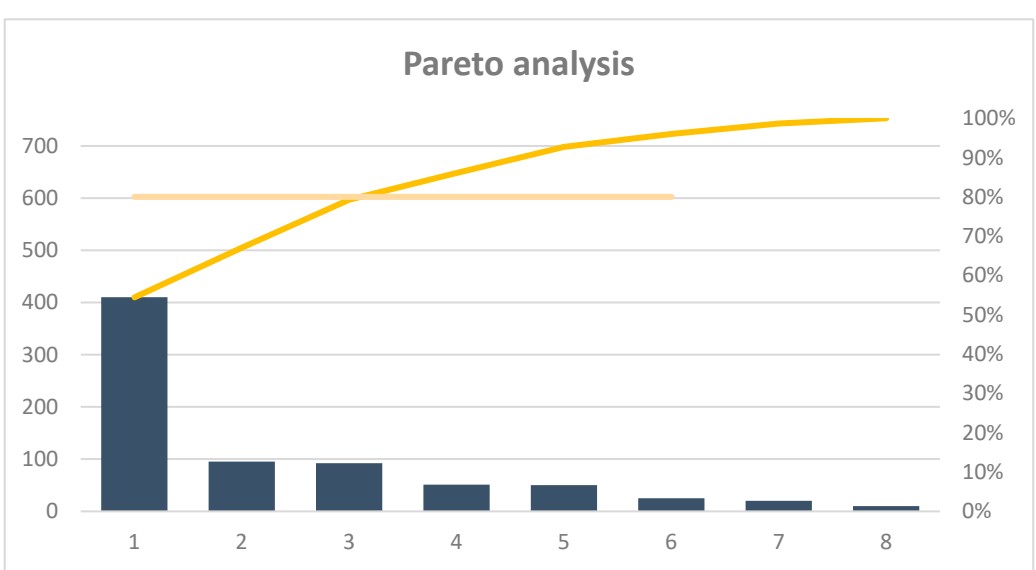

**Figure 2.** Pareto analysis: 1—wrong size; 2—bad forging; 3—technological waste; 4—machine failure; 5—material fallen out; 6—badly cut; 7—burned out; 8—needle pushed.

### 3.2. Implementation of the 8D Report

Implementing a method consists of the following eight (or nine, including D0) steps [3,7,14]: Step D0: Create the problem-solving team. The first step of the 8D report was to create a qualified team with experts from many areas who know the production process in many different ways. The problem-solving team consists of a quality manager, production workers, technicians, production department, logistics department, and maintenance employees.

Step D1: Define the problem. The company accepted the complaint about the product of the Toyota customer. It was the wrong size, according to the bending height detected in several forgings, as shown in Figure 3.

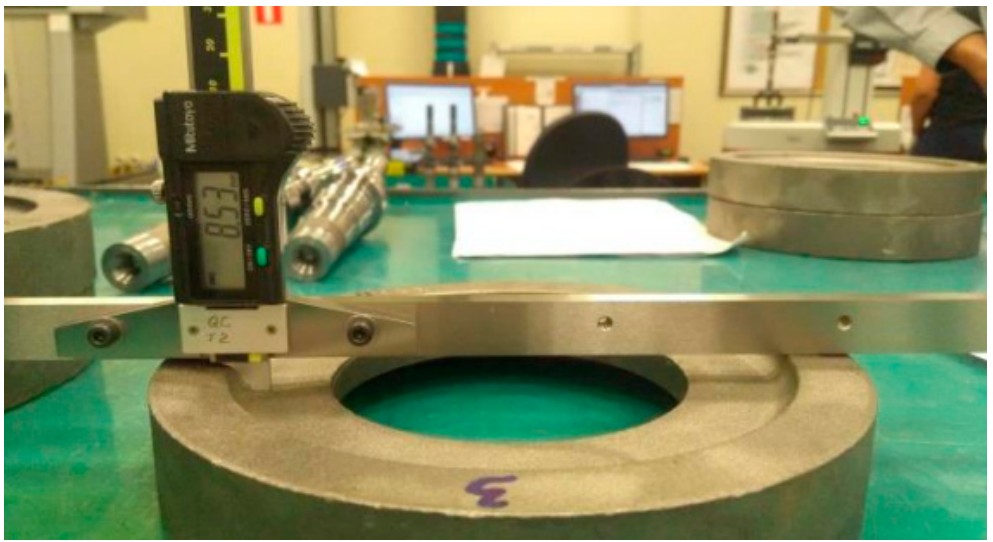

**Figure 3.** Wrong size of the product [15].

Step D2: Description of the problem (is/is not a method). In this step, we can collect facts from production in a company to describe the problem (fault) as accurately as possible, as shown in Figure 4. This information can help us determine the material used in the production process, who the supplier is, how the production process works, which production facility is used in the production process, the maximum tolerance, and the size of the product damage.

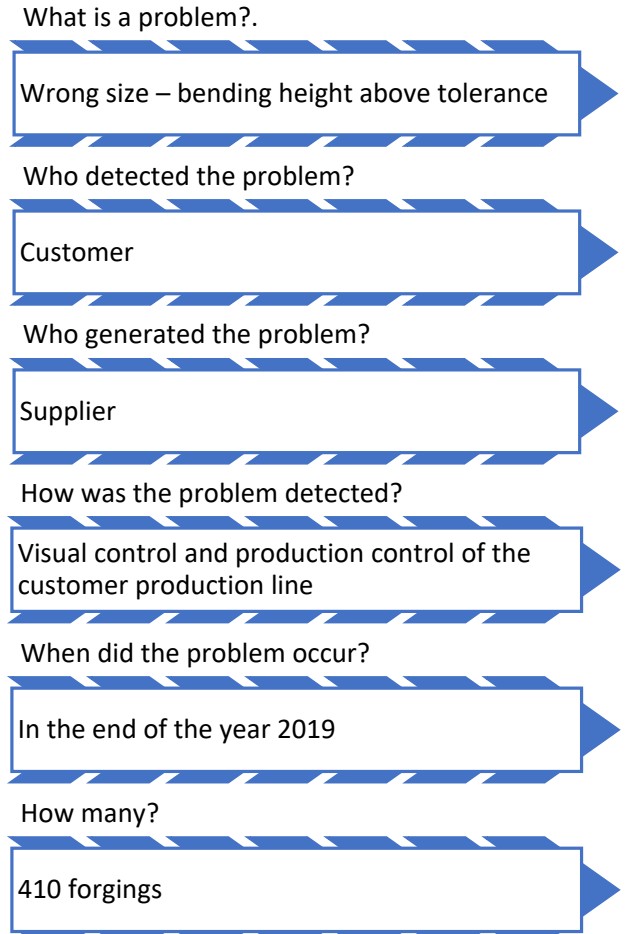

What is a problem?.

Wrong size – bending height above tolerance

Who detected the problem?

Customer

Who generated the problem?

Supplier

How was the problem detected?

Visual control and production control of the customer production line

When did the problem occur?

In the end of the year 2019

How many?

410 forgings

**Figure 4.** Description of the problem [15].

Step D3: Immediate action. After accepting the customer complaint, the semi-finished products in the production process were immediately controlled, and the final products in the production process were stored and controlled. Each product returned to the company has been checked. The staff was immediately trained for better control in the production process. Each worker is informed about the wrong size mistake. Their goal is to observe the causes and circumstances of this mistake's formation. Output control and inter-operational control workers are instructed about this mistake, and their control sheets will be supplemented by a picture of the 3D model sample of the forging (Figure 5).

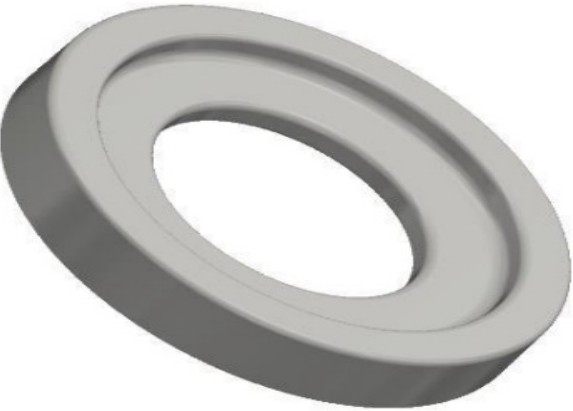

**Figure 5.** Forging 3D model sample [15].

These control sheets for workers will be supplemented with a picture of a simplified component sketch, as shown in Figure 6.

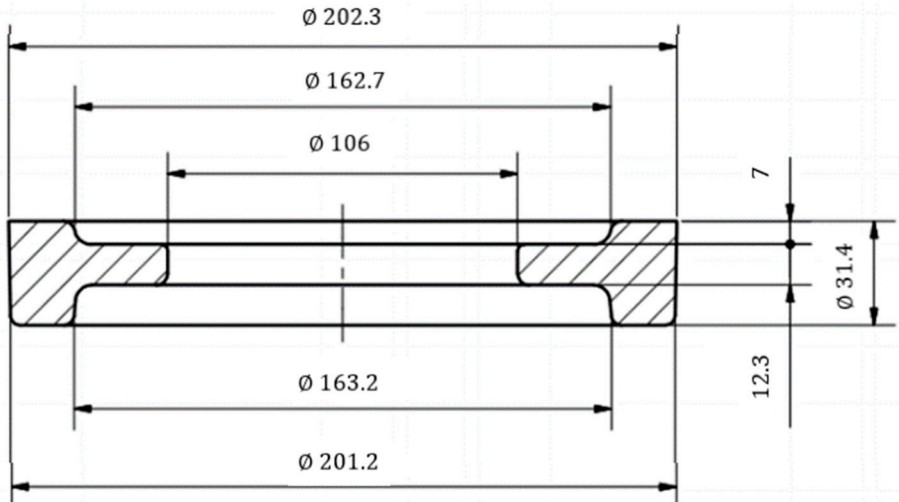

**Figure 6.** Simplified component sketch [15].

Step D4: Identification and verification of root causes. The next step was to identify the possible root causes of the problem by the team. The quality manager, operator, and heads of the production department, logistics department, technics department and maintenance department together identified the problem with the five whys method, and these findings can be seen in Tables 1–3.

**Table 1.** Five Whys Method—Identification of the first root cause [15].

| Probably Trimmed Part of the Cooler on Trimming Machine | |
| --- | --- |
| Cause: Why? | Some parts are bent over tolerance |
| Cause: Why? | The misplaced part on the plate of the trimming machine during the trimming process |
| Cause: Why? | The essential parts were trimmed a little bit colder as usually |
| Cause: Why? | Slower manipulation with the product during the trimming process |
| Cause: Why? | Insufficient attention to trimming from the worker's side |

**Table 2.** Method 5Why—Identification of the second root cause [15].

| Forging | |
| --- | --- |
| Cause: Why? | Slower manipulation |
| Cause: Why? | Longer time for forging |
| Cause: Why? | Insufficient attention for forging from the worker's side |
| Cause: Why? | Higher temperature |
| Cause: Why? | Insufficiently instructed worker with forging process |

**Table 3.** Five Whys Method—Identification of the third root cause [15].

| Control Method | |
| --- | --- |
| Cause: Why? | Wrong parts not found by an operator during the output control |
| Cause: Why? | The output control operator controlled parts according to the control plan |
| Cause: Why? | Insufficient product control during the inter-operational control |
| Cause: Why? | An operator overlooked the wrong size of the product |
| Cause: Why? | All products not inspected |

The first root cause was identified as "probably trimmed part at the coller on trimming machine", as can be seen in Table 1. The team found some parts bent over tolerance,

because the operator misplaced the part on the plate of the trimming machine during the trimming process. The team also discovered that the essential parts were trimmed slightly colder than usually because the operator manipulated the product more slowly during the trimming process. In the fifth why, the team discovered the root cause of the observed problem—insufficient attention to trimming from the worker's side.

As the second root cause was identified as "forging", as can be seen in Table 2. The team discovered slower manipulation, that is, the duration of the forging process was longer. The next discovered problem was insufficient attention to forging from the worker's side, which caused a higher temperature. In the fifth why the team discovered the root cause of the observed problem—worker insufficiently instructed in the forging process.

As the third root cause, the "control method" was identified, as can be seen in Table 3. The team discovered that the wrong parts were not found by an operator during the output control, which means that the output control operator controlled the parts according to the control plan. This was caused by insufficient product control during the inter-operational control. The team also discovered that the operator overlooked the wrong size of the product. In the fifth why, the team discovered the root cause of observer problem—all products were not inspected.

Step 5D: Develop corrective actions. The team established corrective actions and responsibilities after a root cause and effect analysis of the wrong size of the product to prevent future occurrences. It is essential to improve the update and extension of the inter-operation control plan in the production process and to update and extend the output control plan. It is recommended to include random control of products using a sliding scale and more time for employee training. The new improved catalogue of errors was specified.

Step 6D: Verification of corrective actions. The current state was controlled one month after the 8D report process. The verification was carried out by temporary double output control before the product was sent to the customer and retraining the operator. Operators are controlled during the working time, and the production method is controlled.

Step 7D: Preventive actions were used for operator retraining for the trimming machine, forging, inter-operational, and output controls. Operators were retrained for additional measures with a sliding scale by-products control. All operators have a catalogue of errors in their workplace, supplemented by errors in our case.

Step 8D: Give the team credit and define the "Lesson learned". The end of the 8D report gave the team credit. The problem was defined and solved due to the team. Corrective actions were implemented and are set as effective, and, after a few weeks, the number of complaints of this type became lower.

## 4. Discussion

The use of the 8D report in production logistics allows you to effectively analyze all aspects of the problem before concluding and to collect all critical data, whether about the problem or the proposed corrective measures/solutions. All members of the group involved in seeking remedial action are involved in resolving the problem. It is suitable for specific, manageable, and well-defined issues. The advantage is that it plans specific steps to implement and evaluate the recommended corrective action/solution successfully.

The 8D methodology can be considered as a conventional method, but if we combine the 8D methodology with the five whys method and Pareto analysis (in this case), then we can consider this presented solution as synergized and innovative. The framework for a new system standard for product quality improvement is shown in Figure 7. It is a graphical representation of the step sequence for the 8D method in the context of the PDCA cycle and continuous improvement.

The combination and implementation of quality improvement methods and tools was crucial to identify problem, to suggest the solution, and to achieve the continuous improvement that increases the effectiveness of the system. We propose that as a further method of modification, a cause and effect analysis diagram (CE-Ishikawa diagram) can be applied to find the root causes instead of the five whys method. Subsequently, as a

new modification of the presented procedure, Shewhart's control charts could be used to confirm the process stability or process capability evaluation.

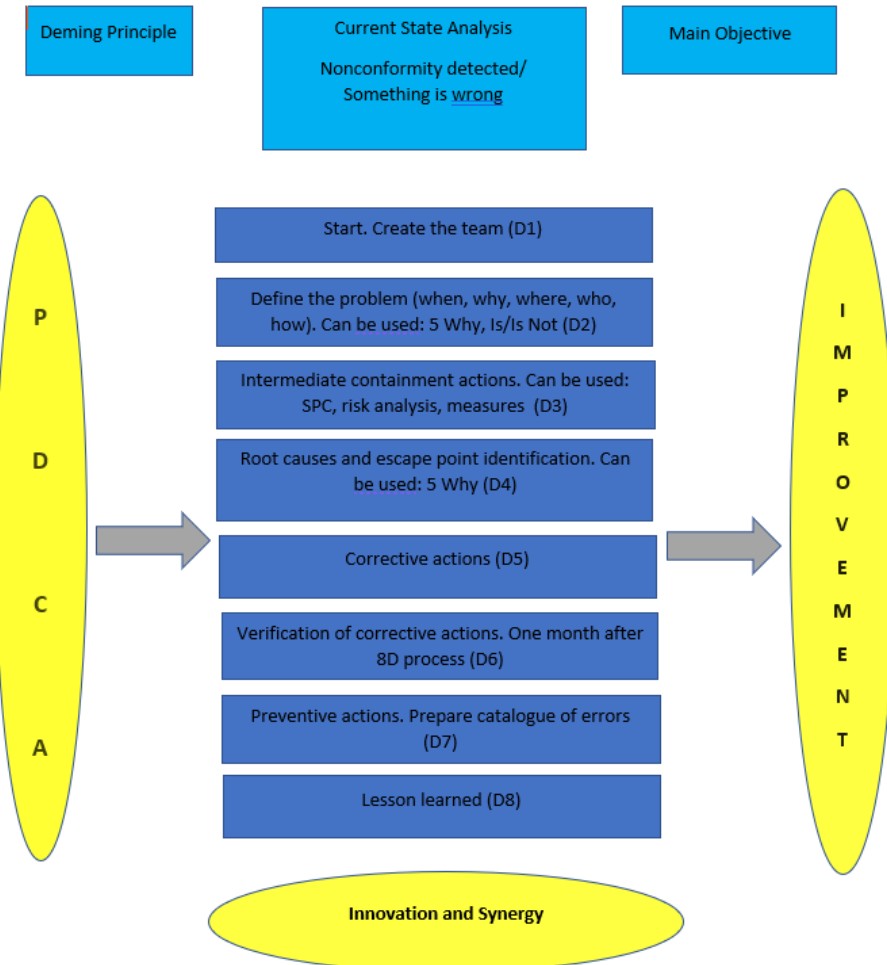

**Figure 7.** The framework for a new system standard for improving the quality of a product.

We consider further that after the corrective actions were implemented, it was essential to monitor the process in the present study, and the forging process was monitored for one month. We focused on the three most important failures—probably trimmed part of the cooler on trimming machine, forging, and control method. The verification was carried out by temporary double output control before the product is sent to the customer, a record of operator retraining was made, operators are controlled during the working time at the working place, and the compliance of the production method is controlled. As the results showed, the number of non-conform products was reduced, representing savings for the organization, and the corrective actions were applied effectively. This confirms that the combination of the introduced methods and tools was appropriate in automotive manufacturing industry.

The 8D report, which focuses on the synergy of interdisciplinary teams and a systematic eight-step approach, can be used to eliminate failures and non-conformities in many industrial sectors, including quality control. Focusing on corrective and preventive measures prevents the occurrence of problems in the future. This method means permanent corrective actions and, by identifying the root causes, prevents their reoccurrence in the future. The 8D method used in the organization and presented in this research is an excellent tool for preventing defects from recurring, as indicated by the PPM (parts per million) results but also by related costs [16].

This is confirmed by main findings of Dziuba et al. [9] in their case study with a similar problem in an enterprise that uses metal stamping technologies to manufacture

products for the automotive and home appliance industries. It concerned the exceeding of length tolerance for the guide section from one hole to another ($43 \pm 0.02$). The 8D report was used to analyze the problem, which identified an important cause of the defect and helped eliminate it. Another case study completed by Alexa and Kiss [17] shows the problems that are occurring with a customer. The customer received a delivery with delivery note114643176, dated 27 June 2015, about which this customer complained, as the packaging contained finished goods that did not comply with its specifications. Improvements implemented in this case study were implemented in the company. The analysis in this case study on quality assurance in the logistics department of the factory ensured the production of quality sensors to prevent defects and ways to avoid recurrence. The third comparable case study was prepared by [18]. The main problem in the company involved gearboxes in various segments, while workers complained to management about poor performance and maintenance issues. The current work provides a motivating perspective on the application of 8D in other manufacturing organizations such as the heavy industry, automobile, and so on. The study has reduced assembly time and defect part numbers from 95 to 26, resulting in a 76.63% reduction in defects. This represents how defects were reduced and time management improved after implementing the 8Ds methodology versus before implementing the 8Ds methodology, demonstrated as well as in a frequency comparison. In this case, the most common defect was a vibrating conveyor, the number of which was reduced from 38 to 10, resulting in a 74 percent success rate.

We can conclude that the implementation of the 8D method is extremely important for improving quality and meeting the needs of customers in the automotive sector. By selecting and modifying this method according to the needs of the specific company and the selected product or process, it is possible to make the production process more efficient. It can be successfully used as part of an innovative way to reduce misunderstandings and the loss of information during the data exchange and for standardizing the process of the information flow in complaint management [19].

**Author Contributions:** The main activities of the team of authors can be described as follows: conceptualization, K.L.Š. and H.F.; methodology, K.L.Š. and H.F.; software, K.L.Š. and A.I.; validation, K.L.Š.; formal analysis, K.L.Š.; investigation, K.L.Š. and M.N.; resources, H.F. and M.N.; data curation, K.L.Š.; writing—original draft preparation, K.L.Š., M.N., A.I. and H.F.; writing—review and editing, K.L.Š. and H.F.; visualization, K.L.Š. and H.F.; supervision, K.L.Š. and H.F. All authors have read and agreed to the published version of the manuscript.

**Funding:** This research received no external funding.

**Institutional Review Board Statement:** Not applicable.

**Informed Consent Statement:** Not applicable.

**Data Availability Statement:** Not applicable.

**Acknowledgments:** This paper was written as a part of this project: KEGA 031STU-4/2020 "Network visualisation of common and specific elements and documented information of integrated management systems concerning relevant ISO standards" with the focus on MSS—ISO 10012:2013 Measurement management systems. Meeting requirements for measurement processes and measuring equipment were supported by the granting agency of The Ministry of Education, Science, Research and Sport of the Slovak Republic and as a part of this project: KEGA 027STU-4/2022 "Integration of the requirements of practise in the automotive industry with the teaching of subjects within the study programs Process Automation and Informatization in Industry and Industrial Management".

**Conflicts of Interest:** The authors declare no conflict of interest.

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
