# Peer review of "Quality Improvement of the Forging Process Using Pareto Analysis and 8D Methodology in Automotive Manufacturing: A Case Study"

_standards, doi:10.3390/standards3010008_

Round 1

Reviewer 1 Report

The paper reports on the application of a method for quality improvement called 8D to a case company.

Although the general topic and the case study are relevant, the paper do not show a clear contribution to the literature neither a clear scientific approach, in the way it is currently written.

What does “it” refer to in line 29?

Last sentence of the introduction is blurry.

First sentence of section 2.1 was repeated in lines 58.

Where does 8D method come from ?

What Pareto analysis is part of in the 8D methodology ?

There is no need to have a table and figure reporting the same data.

English: “8D” should be defined before it is used (in the introduction); “qualities in form as quality”; “monitored organisation is situated”; “exception of customer”; “has been applied, the 8D … has been applied”.

Author Response

What does “it” refer to in line 29? - "it" was deleted

Last sentence of the introduction is blurry. - I agree. (corrected).

First sentence of section 2.1 was repeated in lines 58. - corrected

Where does 8D method come from ? - it was in the text, in introduction and you can find it also in 2. chapter

What Pareto analysis is part of in the 8D methodology ? - Pareto is not a part of some of 8 step, but it hepls us to find which problem is the most important 

There is no need to have a table and figure reporting the same data. - agree, the table is deleted

English - corrected

Reviewer 2 Report

The paper presents a case study of improving production using Pareto Analysis and the 8D procedure. The paper is interesting and can be of interest to the readers. It should be mentioned that the methods discussed aren't new. Multiple papers present similar examples.  That negatively impacts the novelty, scientific soundness and significance of the content. However, I think that the paper is worth publishing.

Author Response

Thank you for the revision.

Reviewer 3 Report

This manuscript (standards-1721896-peer-review-v1) applies the conventional Pareto analysis and 8D methodology in solving the automotive forging product for quality improvement and failure prevention. The research has some practical values in engineering, however I cannot suggest it be accepted in its current form for the following reasons:

1.     The English language throughout the manuscript is not well expressed, e.g. in line 54, it should be ‘… a monitored organization which is situated in …’. Please carefully check the whole manuscript, and it’s better to ask help for a native speaker.

2.     Totally speaking, this manuscript seems like an engineering report rather than a scientific paper, without any deep investigation into materials and mechanics. I don’t know whether it conforms to the requirement of an academic journal.

3.     The methods used are conventional, the only innovation is to apply them in the specific automotive product. Please further elaborate the innovations.

Author Response

Point 1: the manuscript have been checked and corrected

Point 2: I don´t agree, many similar manuscripts can be found about 8D methodology and Standards - special issue if focused on Quality Management Systems.

Point 3: 8D method is 40 years old and used around the world. There is no place to innovate, we can only combine other methods or tools within the 8D, for example Ishikawa or 5 Why to find the root cause, Pareto analysis, maybe Shewhart control charts...

Round 2

Reviewer 1 Report

Minor remarks are addressed. 

Author Response

Dear reviewer,

thank you for your details and insightful comments in reviewing how to improve our paper. We agree with your comment that topic of the paper is relevant to be published in Standards journal.  We significantly improved the paper according to your suggestion; we carefully checked the whole document to improve the English language and style (the changes are marked red). We improve the introduction and discussion.  According to the recommendation we added new cited references of articles published in academic journals and compare the results of the authors with our results. In the part 2.1. Research Aim and Methodology is modified  by adding the more details about research design and decription of the methods used.

We assume that this study can be an example of the specifics of applying quality management tools and methods to achieve the stated goal. Important improvement to meet expectations of reviewers can be find in the discussion part of paper where we discussed in detail that combination of 8D methodology with other methods e.g. Pareto analysis including Lorenz curve and to find the most frequented mistakes and deal with them when proposing measures.

We consider this presented solution as synergy and innovation in process how to improve the quality. The combination and implementation of the quality improvement methods and tools used was crucial to identify the problem, suggest a solution, and to achieve continuous improvement and increase the effectiveness of the system.

The authors of the article would like to thank you very much!

We hope that the improved version of the original paper will be published in the Academic Journal Standards (ISSN 2305-6703).

With best regards authors.

Reviewer 3 Report

I don't think the problems raised in the first round have been adequetely addressed. The decision is up to the editor.

Author Response

Dear reviewer,

thank you for your comments how to improve our paper. According to your suggestion, we carefully checked the whole document to improve the English language (the changes are marked red). We improved the part 2.1. Research Aim and Methodology by adding the more details about research design and decription of the methods used for research and reasons why we choose them.

The aim of the article is to present  study considering improving quality in industry to meet customer expectations.We assume that this study can be an example of the specifics of applying quality management tools and methods to achieve the stated goal.In the introductory part of the research, we focus on choosing the right approach and method by studying the literature.An important task was not only to quantify and focus on the description of failures, but also to focus on the identification causes of failures and to propose measures. Another reason for choosing 8 D method was that it ensures, as mentioned, the effectiveness of control mechanism, while consolidating the use in the future, findings and experience can be used as a basis for future FMEA analysis of a product in automotive.In our opinion , the most important for quality control are that in this case study there were combined many tools of quality management, especially Pareto analysis, RE analysis, 5Why.

Important improvement to meet expectations of reviewers can be find in the discussion part of paper where we discussed in detail that combination of 8D methodology with other methods e.g. Pareto analysis including Lorenz curve and to find out most important / most frequented mistakes and deal with them when proposing measures. Subsequently, the method 5 Why was applied to identify the root causes of the detected failure of product.

We consider this presented solution as synergy and innovation in process how to improve the quality. The combination and implementation of quality improvement methods and tools used was crucial to identify problem, to suggest a solution, and to achieve the continuous improvement and increase the effectiveness of the system. We propose that as a further 8D method modification can be Ishikawa diagram to find the root causes instead of 5 Why, or control charts to confirm the process stability or process capability evaluation. This possibility for future was also discussed in the discussion.

We hope that the improved version of original paper will be published in the academic journal Standards (ISSN 2305-6703). With best regards authors.

Round 3

Reviewer 3 Report

Thanks for your careful modification of your manuscript and sincere response to the round-2 comments. Indeed, from the materials point of view, it is still lack of deep investigation. But in terms of management science, the method is reasonable and the results are acceptable. Considering the journal's scope, particularly this special issue, I can agree to accept it in its current form.

Author Response

Dear Reviewer. We very much appreciate your positive words and thank you for your reply. We agree of course that from the materials point of view, it is still lack of deep investigation. Of course, in the future we will try to pay more attention to the material aspect. As this special issue is dedicated to quality management, we have chosen to focus on the use of quality management tools and methods, especially the 8D method.